



# Rockfall modelling in forested areas: the role of digital terrain model spatial resolution

Barbara Žabota[1], Matjaž Mikoš[2], Milan Kobal[3]

[1] Department of Forestry and Forest Renewable Resources, Biotechnical Faculty, University of Ljubljana, Večna pot 83, 1000 Ljubljana, Slovenia; barbara.zabota@bf.uni-lj.si

[2] Faculty of Civil and Geodetic Engineering, University of Ljubljana, Jamova cesta 2, 1000 Ljubljana, Slovenia;
matjaz.mikos@fgg.uni-lj.si

[3] Department of Forestry and Forest Renewable Resources, Biotechnical Faculty, University of Ljubljana, Večna pot 83, 1000 Ljubljana, Slovenia; milan.kobal@bf.uni-lj.si

*Correspondence to*: Milan Kobal (milan.kobal@bf.uni-lj.si)

**Abstract.** This article examines how digital terrain model (DTM) spatial resolution influences rockfall modelling using a
probabilistic process-based model, RockyFor3D, while taking into account the effect of forest on rockfall propagation and runout area. A rockfall site in the Trenta valley, NW Slovenia, was chosen as a case study. The analysis included DTM spatial resolutions of 1 m, 2 m, 5 m, 10 m, 12.5 m and 25 m, based on lidar data. The highest spatial resolution (1 m) was used to calibrate the surface roughness coefficients of the model while also taking into account the effect of forest since it shapes the rockfall propagation and runout area. The results of the calibration runs were evaluated using goodness-of-fit
indices, and the best set of parameters were further used for modelling rockfalls with and without the effect of forest for all spatial resolutions. Accuracy statistics were used to validate the modelled rockfall propagation and runout area for each spatial resolution, with/without the effect of forest. Finally, modelling outputs, such as the mean of the maximum and maximum kinetic energy, the number of block passes and forest parameters in the rockfall propagation area, were compared.

## Introduction

Mountainous areas are prone to many mass movement processes, rockfalls being one of the most common. Rockfalls can be defined as the separation of rock masses from rock cliffs. This rocky material is transported in various ways, including by falling, bouncing, rolling and sliding (Guzzeti et al., 2002; Petje et al., 2006; Lopez-Saez et al., 2016). Rockfalls represent an important threat to human life and property, and their instantaneous occurrence makes temporal prediction practically impossible (Petje et al., 2006; Abbruzzese et al., 2009). Rockfall models, especially process-based ones, can be an efficient
tool for predicting potential rockfall hazard areas, making it possible to identify rockfall release, transit and deposit areas as



well as to model rock trajectories, the kinetic energy of rocks, rock rebound heights and propagation and reach-out probability (Petje et al., 2005a after Kienholz et al., 1998; Dorren, 2003, 2016). By quantifying the potential rockfall hazard, simulation models can be used for planning different protection measures (e.g. technical measures, natural-based solutions) that can significantly reduce the potential risk of rockfall occurrence in high-threat areas (Dorren et al., 2005; Petje et al.,
2005c; Volkwein et al., 2011; Howlad et al., 2017).

Forests provide a natural solution for protection against rockfalls in alpine regions (Berger et al., 2013; Toe and Berger, 2015) since they can significantly reduce the intensity (kinetic energy) and propagation probability of falling rocks (Dorren and Berger, 2006; Lundström et al., 2009; Bertrand et al., 2013). Forest stands also provide protection against debris flows,
as shown in a case study of an Alpine gorge in Slovenia (Fidej et al., 2015). Although several rockfall modelling approaches have been proposed in the last two decades (e.g. Guzzetti et al., 2002; Crosta et al., 2004; Jaboyedoff and Labiouse; 2011; Christen et al., 2012; Horton et al., 2013; Dorren, 2016), only a few consider the mitigation effect of forest (e.g. Berger et al., 2004; Christen et al., 2012; Dorren, 2016). Models that do consider the protection effect of forest can additionally be used for mapping protection forest and quantifying its protection function against rockfalls (Dorren et al., 2007). In simulation
models (e.g. Christen et al., 2012; Dorren, 2016) the protective effect of forest is considered via the spatial distribution of the forest stand, DBH distribution and tree species. Simulation models enable rockfall hazard mapping with and without the effect of forest, and with different forest scenarios (Dorren et al., 2007; Dupire et al., 2016).

Some rockfall models are intended for use on a local scale (individual rockfall events), while others are more appropriate for
the regional scale (multiple rockfall events) (e.g. Dorren, 2003; Jaboyedoff and Labiouse; 2011; Michoud et al., 2012, Crosta et al., 2015, etc.). The scale of modelling is an important factor when determining the purpose of the modelled results, as it will affect the choice of the spatial resolution of the input data. The most common input data for rockfall simulation models is the digital terrain model (DTM), which carries information about the main morphological properties of the surface. The spatial resolution of the DTM can have a significant impact on modelling both potential rockfall release and runout areas,
primarily on the lateral dispersion of rockfall trajectories (Volkwein et al., 2011; Moos et al., 2018) and on the kinetic energy of rocks (Bühler et al., 2016). In order to achieve the most realistic results, the highest spatial resolution should be used in rockfall modelling; however, since individual rockfall models are designed for use at a particular spatial level, higher spatial resolution does not necessarily also provide improved modelling results (Zieher et al., 2012). Although a few studies have investigated the influence of changing DTM spatial resolution on rockfall modelling (e.g. Loye et al., 2009; Lan et al., 2010;
Zieher et al., 2012; Frattini et al., 2013; Bühler et al., 2016; Žabota et al., 2019), none have also focused on analysing the protection effect of forest.

Accordingly, this paper analyses the impact of changing spatial resolution on the protective effect of forest in rockfall propagation and runout areas. The aim was to quantify the effect of forest on rockfall runout by taking into account several



DTM spatial resolutions and to observe how it influences rockfall hazard assessment. For modelling rockfall propagation area, the RockyFor3D model was used (Dorren, 2016), and modelling was performed at different DTM spatial resolutions (1 m, 2 m, 5 m, 10 m, 12.5 m, 25 m). The model was applied with and without the effect of forest, while forest stand conditions remained the same. Based on this analysis we investigated i) the effect of DTM spatial resolution on the modelling of rockfall propagation and runout areas, ii) the extent to which forest reduces rockfall propagation at different DTM spatial

resolutions and iii) the influence of forest on various model output parameters at different DTM spatial resolutions.

## 2 Materials and methods

### 2.1 Study area

The modelled area was a rockfall site located in the Trenta valley in Triglav National Park in the NW part of Slovenia (Fig. 1). The last large rockfall event occurred in April 2017, when approximately 29,400 $m^3$ of coarse material was released. The

rockfall area and the surrounding area are composed of Upper Triassic layered limestone (SFRJ, 1986). In the rockfall deposit area, it was possible to detect rockfall deposits that resulted from older rockfall activity, as indicated by the colour of the rocks (older rocks are darker) and by vegetation cover (older rocks are overgrown by low shrubs and dwarf pines). Similar findings were also obtained when comparing orthophoto images from different years. In this study only the newest rockfall event, which comprises 19,342 $m^2$, was modelled.


### 2.2 Rockfall modelling

The RockyFor3D rockfall model (Dorren, 2016) was used for modelling rockfall propagation and runout area. RockyFor3D is a probabilistic, process-based rockfall trajectory model of falling blocks in three dimensions (Dorren, 2016) that can be used for regional, local and slope-scale rockfall simulations. Rockfall trajectory is simulated as 3D vector data by calculating

sequences of classical parabolic free falls through the air, rebounds on the slope surface, and also impacts against trees (optional). In the model rolling is represented by a sequence of short-distance rebounds, while rock sliding is not modelled. The required input data include the topography and surface slope characteristics, as well as a set of parameters that define the release conditions. The minimum input data required include the DTM, definition of the release area, rock density, rock size and shape, surface roughness and soil type (Dorren, 2016). RockyFor3D also enables simulation with forest, which can be

done either i) by providing a text file with locations of the trees, their stem diameter at breast height (DBH) and the percentage of coniferous trees, or ii) by providing four raster maps containing the number of trees, mean DBH, standard deviation of DBH and the percentage of coniferous trees. Additionally, the model enables simulation with rockfall nets as protection structures on a slope. The main outputs of the model are maximum kinetic energy (90 % confidence interval of all maximum kinetic energy values), maximum bounce height, the number of block passes through each cell, rockfall



propagation probability, the number of deposited blocks, maximum simulated velocity, maximum tree impact height and the number of tree impacts per cell (Dorren, 2016).

## 2.3 Used datasets

The input data that was of interest in this study was the DTM. In the guidelines for use of the RockyFor3D model, it is stated

that both the spatial precision of the simulated maps and the accuracy of the simulated kinematics decrease with increasing cell size (Dorren, 2016). However, the experience of the authors is that a 1×1 m spatial resolution does not necessarily improve the modelling results, and that the preferred spatial resolution lies between 2×2 m and 10×10 m (Dorren and Heuvelink, 2004). In order to test the preferred spatial resolution of the data, we used the following set of DTM spatial resolutions: 1×1 m (DTM1), 2×2 m (DTM2), 5×5 m (DTM5), 10×10 m (DTM10), 12.5×12.5 m (DTM12.5), and 25×25 m

(DTM25). DTM data were created based on 1×1 m lidar data (SMARS, 2014) obtained in 2014, three years before the last rockfall event in the study area.

By comparing the point clouds before (lidar point cloud) and after the rockfall event (photogrammetric point cloud obtained from the UAV survey in July 2018), and based on comparison of orthophoto images, we were able to determine the rockfall

release area, which encompasses approximately 4,007 m². Other initial inputs and parameters for the simulation model were determined based on a field survey and UAV observations. The rock density was set to 2500 kg/m³ (limestone release area; Berger and Dorren, 2007), and, based on the height difference between release and deposit areas, the initial fall height was set to 50 m. Block dimensions and block sizes were determined using data from 368 rocks that were deposited at the maximum runout of the rockfall. Consequently, the calculated block dimensions used for modelling were 1.4 m, 0.9 m and

0.8 m, respectively, while the prevailing block shape was rectangular. Variation in block volume was based on field observations and set to ± 50 %.

Surface roughness parameters (rg) represent rocks lying on the slope that form obstacles for falling rocks (Dorren, 2016). The parameters define the surface roughness, which is expressed as the size of the material covering the slope's surface in

the downward direction of the slope. Rg70, rg20 and rg10 correspond to 70 %, 20 % and 10 % of the cases during a rebound on the slope, respectively, and represent values from 0 to 100 (0 represents a smooth surface). In the first simulation rg parameters were set to rg70 = 0.25, rg20 = 0.5, rg10 = 0.9 (talus slope with average block diameter larger than 10 cm); however, in order to calibrate the model, a range of rg values were used in the calibration process (see Chapter 2.4). Soil type was chosen according to the guidelines of the model (talus slope with Ø > ~10 cm, or compact soil with large rock

fragments). The number of simulations was set to 1000, as recommended by the authors. The modelling was also performed with and without the effect of forest. Simulation with forest was done by using tree locations, stem diameter at breast height





(DBH) and percentage of coniferous trees. Tree locations and additional attributes were obtained by using a digital surface model (DSM) and DTM1 using lidar data from 2014.

**2.4 Calibration of surface roughness parameters**

The choice of surface roughness parameters (rg70, rg20, rg10) must be done with care, as the model is sensitive to these parameters (Bourrier and Hungr, 2013; Gischig et al., 2015; Dorren, 2016). Therefore, calibration of these parameters was done using a range of values that changed at the same time and the same rate. The combination of rg values used for calibration was 0.15, 0.4, 0.8 – 0.26, 0.51, 0.91. Calibration of the model was performed using the highest spatial resolution (DTM1), and by taking into account the effect of forest.


For evaluating the performance of each calibration run, we used goodness of fit indices (GOF) after Formetta et al. (2016). GOF indices are based on pixel-by-pixel comparison between the observed rockfall area map (OR) and predicted rockfall area map (PR) (Table 1). Comparison of these two maps results in binary maps with positive values corresponding to "actual rockfall area" and negative values corresponding to "not a rockfall area". Correspondingly, four types of pixel outcomes are

possible for each raster cell: i) true positive (TP) is a pixel mapped as an "actual rockfall area" on both the OR and PR (correct prediction), ii) true negative (TN) is a pixel mapped as "not a rockfall area" on both the OR and PR (correct detection of areas where rockfalls do not occur), iii) false positive (FP) is a pixel that is actually "not a rockfall area" on the OR but is mapped as an "actual rockfall area" on the PR (false alarm), and iv) false negative (FN) is a pixel that is an "actual rockfall area" on the OR but is mapped as "not a rockfall area" on the PR (missed alarm) (summarized based on Formetta et

al., 2016). These indices are the basis of the concept of receiver operator characteristics (ROC; Goodenough et al., 1974) that are used for assessing the model performance using the relation between benefits (TP) and costs (FP). Formetta et al. (2016) incorporated eight GOF indices in the ROC system for quantification of model performance; however, four indices have been shown to be the most suitable for evaluation of calibration runs (Table 2): the success index (SI), distance to perfect classification (D2PC), the average index (AI), and true skill statistics (TSS). More comprehensive and detailed descriptions

of the indices are available in Formetta et al. (2016).

The calibration run that achieved the most optimal value with those indices was selected as the most successful, and the rg values of that calibration run were used for modelling rockfall runout area at all spatial resolutions considering two modelling scenarios: with and without forest. For the modelled rockfall area, the output propagation probability was used

since this raster layer represents the most realistic spatial distribution of the current rockfall event and can be used for calculating spatial occurrence probability, which is used in rockfall hazard analyses (Dorren, 2016).

**2.5 Validation of the modelling results**





Validation of rockfall runout zones for different spatial resolutions and both modelling scenarios was done based on
accuracy statistics derived from the confusion matrix (Beguería, 2006) by using statistics that are not dependant on
prevalence. The following accuracy statistics were used in this study to validate the model performance (presented in Table
3): sensitivity, specificity, false positive rate and false negative rate. In order to support the validation procedure, additional
outputs of the simulation model were analysed and argued in the results, namely: the surface area and runout lengths, the
mean of the maximum kinetic energy, the maximum energy value recorded in a given cell, and the number of block passes
through each cell. Additional features related to the effect of forest in the modelling that were compared were the number of
trees in the propagation area, which changes when modelling with the effect of forest, average DBH values of those trees,
and the mean of the maximum kinetic energy reached at the locations of those trees.

## 3 Results

### 3.1 Model calibration for DTM1

Model performance was evaluated using the set of data for the DTM1 spatial resolution (with the effect of forest) and a
combination of 12 values of surface roughness coefficients (rg). The results of calibration runs of the GOF indices used are
presented in Table 4. The results show that TPR values decrease, while FPR values increase, with increasing rg values. The
most successful model performance can be attributed to calibration run 01, as the difference in the FPR rate is lower
compared to the difference in TPR rate between the best (01) and worst (12) model run (Fig. 2). As rg values increase with
each calibration run, the modelled extent of the rockfall runout zone decreases, resulting in greater underestimation of the
actual rockfall propagation area. Looking at the remaining GOF indices in Table 5, it can be observed that the index values
of all model runs are similar and do not deviate by more than 0.2. The larger deviation in values is achieved in the case of
D2PC, where it can be observed that by increasing rg values, the D2PC increases. Considering all GOF indices presented in
Table 5, it can be concluded that the best model performance is achieved by calibration run 01 ($rg70 = 0.15$, $rg20 = 0.4$, $rg10$
$= 0.8$).

### 3.2 Validation of the modelled rockfall runout zones with and without forest

The results of sensitivity and specificity statistics show (Table 5) that differences in the prediction rate between the correctly
predicted positive and negative rates are minimal in the modelling of rockfall runout areas, both with and without the effect
of forest as well as for different DTM spatial resolutions. The sensitivity rate is higher when the effect of forest is not taken
into account in the modelling process for the majority of spatial resolutions, while the results of the specificity rate show that
better prediction of negative cases is achieved when forest is taken into account in the modelling of rockfall propagation area
(except DTM12.5 and DTM25 in both cases). The highest sensitivity rate is achieved by DTM1 in both modelling scenarios,
while the lowest is achieved by DTM5 in the forest scenario and DTM12.5 in the scenario without forest. The highest value



as well as the lowest specificity rate is achieved by DTM5. There is no clear trend in relation to decreasing DTM spatial resolution and forest modelling scenarios when observing positively predicted rockfall runout area.

In both modelling scenarios FPR rate is the largest in the case of DTM1, while the lowest rate is in the case of the no forest scenario achieved by DTM12.5 and in the forest scenario of DTM5. FPR values decrease with decreasing DTM spatial

resolution in both modelling scenarios, while FNR values increase, except at DTM12.5 and DTM25, which have the same FNR in both scenarios. The majority of spatial resolutions have a lower FPR rate when forest is not taken into account in the modelling process, and higher FNR values when forest is taken into account. DTM1 achieves the lowest FNR values in both modelling scenarios. In the case of these modelling results, the FNR rate is an indicator of the largest differences and of when the model wrongly predicts areas that are not part of the rockfall runout area in the model.


The rockfall propagation and runout area decreases with decreasing DTM spatial resolution in both modelling scenarios (Fig. 3), and the model overestimates the actual rockfall extent at all spatial resolutions. The largest overestimation of the propagation area in all spatial resolutions and scenarios is in the northern part of the rockfall propagation area (lateral part of the rockfall source area) and in the maximum rockfall runout, while the southern lateral side exhibits the greatest match

between the modelled results and actual rockfall outline. On average, the largest overestimation of rockfall propagation area occurs when forest is not taken into account, namely at DTM1 and DTM5 (35.3 m and 35.6 m). Underestimation of the propagation area is similar in both modelling scenarios; the lowest values in both are at DTM1. When observing the shape and extent of the runout area of the modelled results, it can be concluded that as DTM spatial resolution decreases, the rockfall propagation area becomes more generalized as the surface topography is disregarded, and the shape of rockfall

propagation area is not represented accurately. The same can be concluded for both the forest and no forest scenarios. When observing the forest and no forest modelling results, it can also be stated that for DTM1, DTM2 and DTM5 spatial resolutions, the shape of the forest scenario changes according to the no forest scenario, and the change is only observed for the maximum runout distances. On the other hand, at DTM10, DTM12.5 and DTM25 spatial resolutions, it can be observed that the difference between the forest and no forest scenarios is not as clear as at the other spatial resolutions – both in the

shape and maximum runout zone. DTM1, DTM2 and DTM5 spatial resolutions also provide comparable results with respect to the shape of the rockfall propagation, while modelling results of DTM10, DTM12.5 and DTM25 spatial resolutions are not comparable with the others.

The forest has the largest impact on rockfall propagation area at DTM2, reducing it by 24 %, followed by DTM1 (19 %) and

DTM5 (13 %) (Fig. 4). In the case of DTM25, the rockfall propagation extent is the same in both modelling scenarios, while the impact of forest is low in the case of DTM10, where the propagation area was only reduced by 5 %. In the case of DTM12.5, the rockfall propagation area is larger when forest is taken into account (by 4 %). When forest is included in the simulation, the largest overestimation of rockfall propagation is on average achieved by DTM25 (16.9 m), while DTM5 (7.9



m) exhibits the least overestimation on average. The underestimation rate of the simulation model is lower than the overestimation rate; the model underestimates the most at the maximum runout area (based on the location of the source area). The largest change in average rockfall runout zone length for modelling scenarios with and without forest is at DTM2, where the average runout zone decreased from 35.6 m to 10.6 m, followed by DTM1 (35.3 m → 13.7 m) and DTM5 (22.1 m → 7.9 m). At DTM10 the length decreased only by 1.3 m. At DTM12.5 and DTM25 the average length increased by 5.5 m and 2.9 m, respectively.


### 3.3 Comparison of model outputs according to the forest / no-forest scenarios

The results of different parameter outputs of the model are summarized in Table 6. The maximum energy value recorded in each cell (Ph_95CI) decreases with decreasing DTM spatial resolution in both the forest and no-forest scenarios. The greatest drop in kinetic energy is between the DTM1, DTM2 and DTM5 spatial resolutions for the forest scenario, while the 240 differences between the other spatial resolutions are smaller. In no-forest scenarios, the differences in the maximum mean kinetic energy are smaller. The most evident change between modelling scenarios is at DTM1 and DTM2, where the maximum value decreases by 40 % and 58 %, respectively, while at other resolutions it even increases by a small percentage (< 2%). The mean of the maximum kinetic energy (E_mean) at DTM1 and DTM2 is larger when forest in taken into account, namely by 128.31 kJ at DTM1 and 437.14 kJ at DTM2.


The maximum number of deposited rocks in one raster cell is higher when forest is taken into account in the modelling. This is because the maximum runout zone is less spatially extensive than when forest is not taken into account, and deposits in the model are more channelized. The maximum number of rock passages through one cell is lower when forest is part of the modelling. The largest differences between modelling scenarios with and without forest are exhibited in both the number of 250 deposited rocks and rock passages only at DTM1, where the effect of forest has the biggest influence on rockfall runout zone.

The number of trees that are located in the area, which is reduced by the impact of the forest, is the largest at DTM5 (255) and DTM1 (254) (Table 7). This is one of the reasons that the effect of forest is the largest at these two spatial resolutions, as 255 the modelled propagation area without forest has more than 54 % more trees compared to other spatial resolutions. Average DBH values of the trees that were in the area, which was reduced due to the protection effect of forest, vary between spatial resolutions. The highest average DBH value was achieved by DTM12.5 (41.81 cm), followed by DTM2 (39.39 cm) and DTM1 (39.17 cm), while the lowest average DBH values were at DTM10. More than half of trees located in the area that was decreased due to the effect of forest had DBH values larger than 30 cm. The mean of the maximum kinetic energies of 260 modelled rocks in the area of those trees varied between spatial resolutions; the largest was at DTM2 with 422.5 kJ, followed by DTM5 (352.7 kJ), while DTM10 (194.63 kJ) and DTM12.5 (173.037 kJ) achieved the lowest values.



## 4 Discussion and conclusions

In this study we investigated the influence of DTM spatial resolution and the protection role of forest on modelling rockfall propagation and runout area. While setting up the modelling environment, we firstly calibrated the simulation model with rg values using GOF indices (Formetta et al., 2017) based on DTM1 with the forest scenario. The challenging part of using a larger number of GOF indices is how to combine their values into common indices in order to evaluate the model performance since all indices do not clearly point to only one model run and are not consistent with changing input parameter values. Based on the values of GOF indices, D2PC is the most consistent and, compared to other indices, is able to better capture the real variations of the changing input parameters (Formetta et al., 2017). An additional challenge that we encountered was the use of the optimal combination of rg parameters since the calibrated result still does not provide the optimal simulation of propagation area. The results are still comparable with the actual state of the rockfall propagation and runout area (e.g. Trappmann et al., 2014), but perhaps the calibration could be improved if i) the rg values were not changed simultaneously, ii) more combinations of coefficients were used, iii) the calibration was done together with soil types (Corona et al., 2017), and iv) calibration was done for an individual spatial resolution. In future rockfall modelling we should use new techniques for determining surface roughness coefficients and other geometrical information on endangered slopes and falling rocks, e.g. UAV technology (e.g. Saroglou et al., 2018; Vanneschi et al., 2019), which could improve the calibration of both surface and soil parameters and reduce the amount of field work.

DTM1-DTM5 spatial resolutions have proven to predict the shape of the rockfall area the most accurately, while at DTM10-DTM25 the shape of the propagation area is simplified and it indicates the shape of raster cells (Zieher et al., 2012). Overestimation is greatly reduced when the effect of forest is a part of the rockfall simulation at DTM1-DTM5 spatial resolutions since the runout zones of those DTMs have the longest runout distances and actually reach a larger forest area. The effect of forest is not evident at DTM10-DTM25 spatial resolutions. The influence of forest on modelling rockfall propagation area is most pronounced with respect to FPR and FNR rates. The FPR rate achieves the best results at lower spatial resolutions (due to larger overestimations at higher resolutions), while the FNR rate achieves the best modelling result with higher spatial resolutions, as they are better able to predict the actual rockfall area. To estimate the best overall performance of the spatial resolution and model scenarios, it is important to consider the differences in the accuracy statistics values, especially as they do not provide a clear solution on which model run is actually the most successful. This means that these two rates should indicate which model (spatial resolution) has better performance. Considering only these two rates and the fact that the rates have better results when forest is included in rockfall modelling, it can be concluded that in both cases DTM1 achieves better results, followed by DTM10 and DTM2.

Forest not only reduces the rockfall propagation and runout area at all spatial resolutions, but also the maximum kinetic energy (e.g. Rammer et al., 2015; Bühler et al., 2016; Lopez-Saez et al., 2016; Moos et al., 2017). However, the maximum





kinetic energy can be greater and rock rebounds higher in some parts of the slope with the effect of forest, which can be explained by the forest's channelizing effect on rocks. The average DBH of trees that were located in the area that was reduced by the protection effect of forest differs between spatial resolutions, which is related to the number and locations of those trees. DTM1-DTM5 spatial resolutions cover the area of compacted forest while others have shorter runout distances that only reach the forest edge where trees with lower DBH values are located. Tree data that were included into the

simulation model were extracted from lidar data. An issue that we encountered is that due to errors, the height and DBH of trees are often overestimated, which can affect the final output result. Moreover, in areas of higher tree density, not all trees are identified, and the number of missing trees is not known in the simulation model (Monnet and Bourrier, 2014). The issue that needs to be considered regarding the inclusion of tree species in the model is the percentage of coniferous trees, as this could be critical for better rockfall propagation assessment (Monnet and Bourrier, 2014; Moos et al., 2018). At lower spatial

resolutions the use of average conifer percentage might lead to incorrect computation of species to the diameter class.

The findings of this study confirm that forest should be considered as part of rockfall hazard assessment (e.g. Clouet et al., 2012; Dupire et al., 2016; Moos et al., 2018). Combining rockfall simulation tools with forest can lead to the identification of areas where the protective effect of the forest is insufficient and where additional technical protection measures should be

considered (Radtke et al., 2014; Duipre et al., 2016; Moos et al., 2017, 2018). This data can help in planning the location and capacity of the required protection measures (e.g. the height of the nets and their capacity) and the associated costs (Fuhr et al., 2015; Dupire et al., 2016). Still, in order to appropriately evaluate the protection effect of forest, more detailed forest information should be included into the rockfall simulation model given that the current version of RockyFor3D only distinguishes conifers from broadleaved trees (Dorren, 2016). This could include tree species data, the vitality of trees, tree

anchoring and site specific characteristics (Moos et al., 2018).

Nevertheless, the decision on the protective effect of forest and additional protection measures will be strongly correlated with the DTM spatial resolution. The protection effect of forest was the most accurately portrayed at DTM1-DTM5, while the other spatial resolutions are too coarse to be used in planning of other protection measures. It is also important to state

that a high DTM spatial resolution might be very sensitive to small changes in elevation, which can be observed in the rockfall simulation model results as high fluctuation/variability of e.g. maximum kinetic energy or bounce heights in a small area. This means that the results of the simulation model should not be applied blindly but with great care. Providing safe conditions for human activities must always come first. Thus, the decision on DTM spatial resolution must be consistent with the main goal of the final modelling results. Based on the results of this study, the preferred spatial resolutions on the

local scale are DTM1, DTM2 and DTM5, while the others do not provide results that are realistic enough.

*Data availability.* Co-authors used publically available lidar data for Slovenia, provided by The Surveying and Mapping Authority of the Republic Slovenia (http://gis.arso.gov.si/evode/profile.aspx?id=atlas_voda_Lidar@Arso).



*Competing interests.* The authors declare that they have no conflicts of interest.

*Acknowledgements.* This research was funded through the Interreg Alpine Space projects ROCKtheALPS (ASP 462) and GreenRisk4ALPs (ASP 635).

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





**FIGURE CAPTIONS**

**Figure 1.** Location of the study area in the NW part of Slovenia with an aerial image of the rockfall.

**Figure 2.** Comparison of calibration run 01 and 12 for DTM1 with taking into account the effect of forest. It can be observed that for calibration run 01 (the lowest rg values), the model has better results at the maximum runout zone, while at the northern part (right lateral side from the source area) the model result are poorer for calibration run 01.

**Figure 3.** Modelling results for DTM spatial resolutions (1, 2, 5, 10, 12.5, 25 m) taking into account two modelling scenarios: with and without the protection role of forest.

**Figure 4.** Modelled rockfall propagation area in relation to changing DTM spatial resolution (1, 2, 5, 10, 12.5, 25 m) and two modelling scenarios (with and without the protection role of forest). Rockfall propagation area (m2) is the largest in DTM1 and decreases with the use of a coarser DTM. Simulation results show that the propagation area is larger without forest. At DTM12.5 and DTM25 the propagation area is almost the same in both scenarios (forest / no-forest).

**TABLE CAPTIONS**

**Table 1.** Confusion matrix showing the relation between observed and predicted rockfall area: TP – true positive, FP – false positive, TN – true negative, FN – false negative.

**Table 2.** Indices of goodness-of-fit (GOF) for comparison between actual rockfall area and predicted rockfall area, after Formetta et al. (2016).

**Table 3.** Accuracy statistics not dependant on prevalence (Beguería, 2006), used for validation of the modelling results.

**Table 4.** GOF index values, AI, DP2D, SI and TSS, for the calibration run using DTM1 and taking into account the effect of forest.

**Table 5.** Validation results for modelling scenarios taking into account different spatial resolutions and two modelling scenarios: with and without the effect of forest.

**Table 6.** The maximum values of output rasters of the RockyFor3D model for scenarios with and without forest: the mean of the maximum kinetic energy in kJ (E_mean), the maximum energy value recorded in a given cell/the 95% confidence interval of all maximum kinetic energy values in each cell in kJ (Ph_95CI), the number of blocks stopped in each cell (Nr_d), and the number of block passes through each cell (Nr_p).

**Table 7.** The number of trees in the propagation area that changes when modelling with the effect of forest (Nr_trees), average DBH values of those trees (DBH average) and the mean of the maximum kinetic energies (E_mean) reached at the locations of those trees.





530

**FIGURES**

**Figure 1.** Location of the study area in the NW part of Slovenia with an aerial image of the rockfall.

535

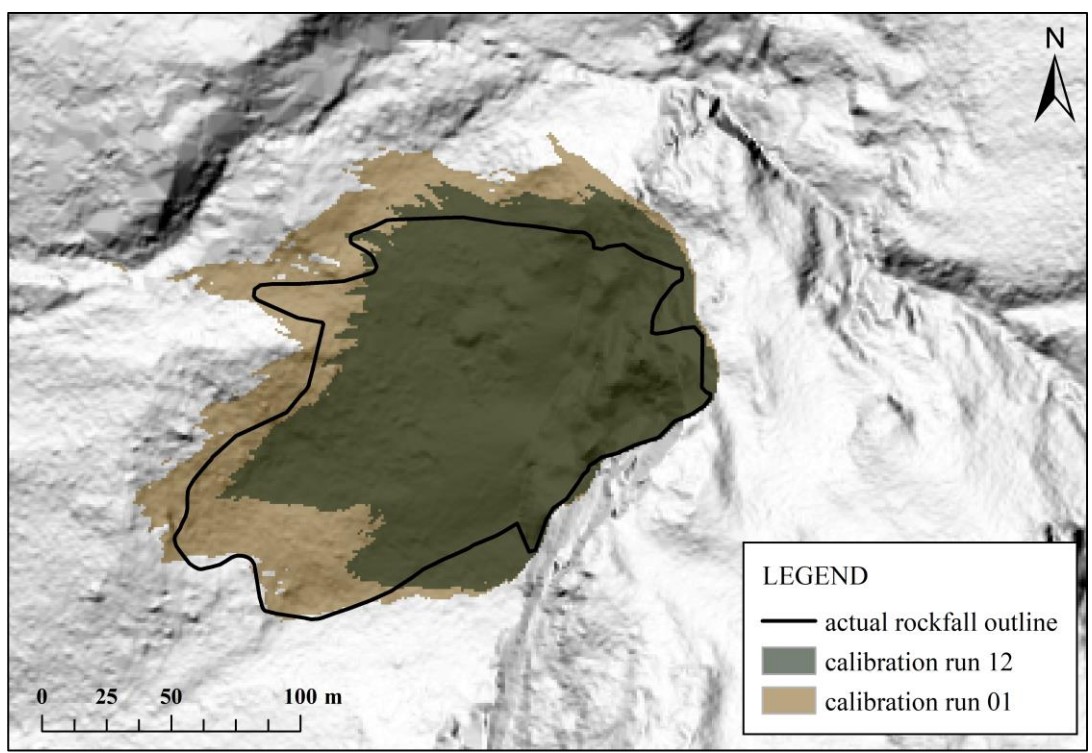

**Figure 2.** Comparison of calibration run 01 and 12 for DTM1 with taking into account the effect of forest. It can be observed that for calibration run 01 (the lowest rg values), the model has better results at the maximum runout zone, while at the northern part (right lateral side from the source area) the model result are poorer for calibration run 01.


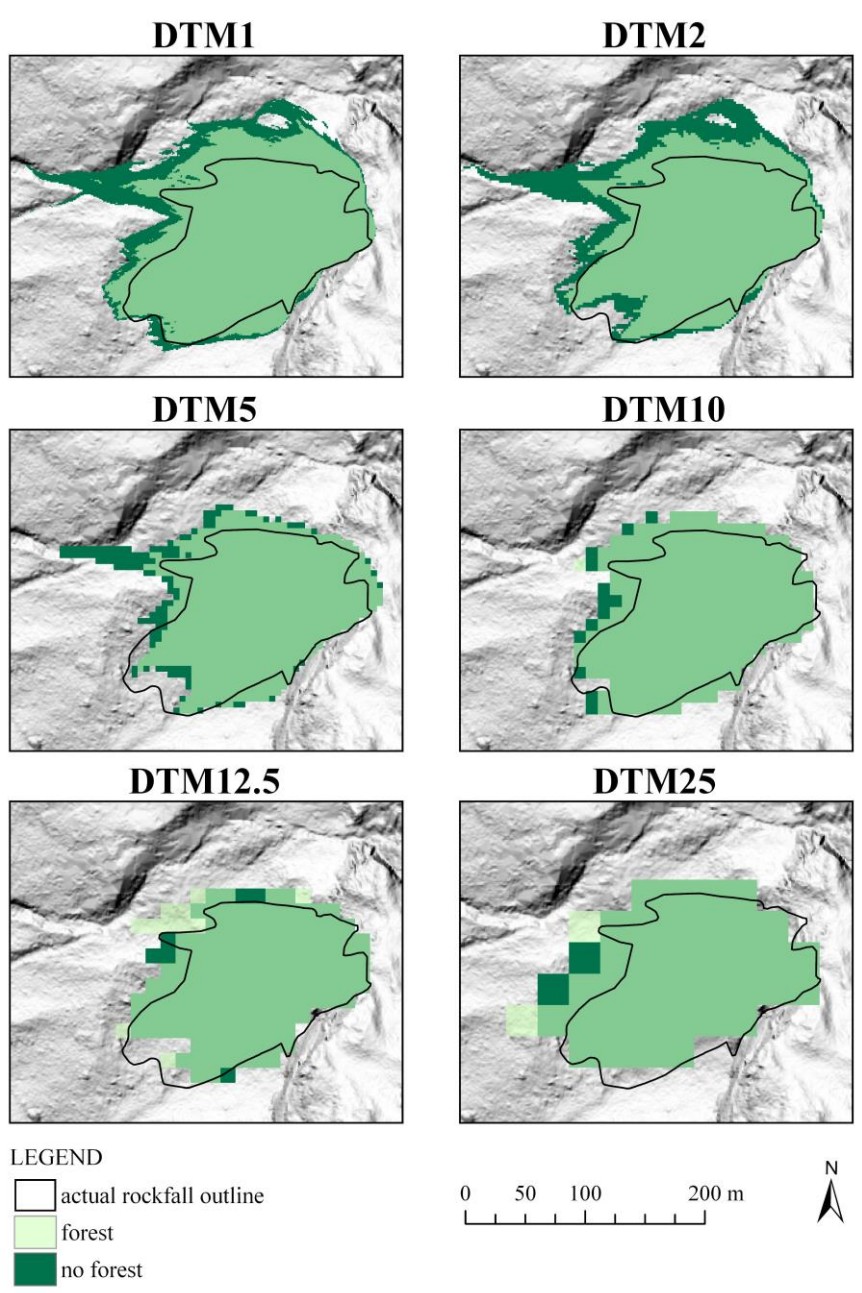

**Figure 3.** Modelling results for DTM spatial resolutions (1, 2, 5, 10, 12.5, 25 m) taking into account two modelling scenarios: with and without the protection role of forest.





**Figure 4.** Modelled rockfall propagation area in relation to changing DTM spatial resolution (1, 2, 5, 10, 12.5, 25 m) and two modelling scenarios (with and without the protection role of forest). Rockfall propagation area (m2) is the largest in DTM1 and decreases with the use of a coarser DTM. Simulation results show that the propagation area is larger without forest. At DTM12.5 and DTM25 the propagation area is almost the same in both scenarios (forest / no-forest).





**TABLES**

**Table 1.** Confusion matrix showing the relation between observed and predicted rockfall area: TP – true positive, FP – false positive, TN – true negative, FN – false negative.

|  | observed rockfall | |
|---|---|---|
| **predicted rockfall** | **X₁** | **X₀** |
| **X'₁** | TP | FP |
| **X'₀** | FN | TN |

560

**Table 2.** Indices of goodness-of-fit (GOF) for comparison between actual rockfall area and predicted rockfall area, after Formetta et al. (2016).

| Name | Definition | Range | Optimal value |
|---|---|---|---|
| Success index (SI) | $SI = \dfrac{1}{2} \times \left( \dfrac{TP}{TP + FN} + \dfrac{TN}{FP + TN} \right)$ | **[0, 1]** | **1.0** |
| Distance to the perfect classification (D2PC) | $D2PC = \sqrt{(1 - TPR)^2 + FPR^2}$ | **[0, 1]** | **0.0** |
| Average index (AI) | $AI = \dfrac{1}{4} \times \left( \dfrac{TP}{TP + FN} + \dfrac{TP}{TP + FP} + \dfrac{TN}{FP + TN} + \dfrac{TN}{FN + TN} \right)$ | **[0, 1]** | **1.0** |
| True skill statistics (TSS) | $TSS = \dfrac{(TP \times TN) - (FP \times FN)}{(TP + FN) \times (FP + TN)}$ | **[-1, 1]** | **1.0** |

**Table 3.** Accuracy statistics not dependant on prevalence (Beguería, 2006), used for validation of the modelling results.

| Name | Definition | Explanation |
|---|---|---|
| Sensitivity (SEN) | $SEN = \dfrac{TP}{TP + FN}$ | the proportion of positive cases correctly predicted |
| Specificity (SPE) | $SPE = \dfrac{TN}{TP + TN}$ | the proportion of negative cases correctly predicted |
| False positive rate (FPR) | $FPR = \dfrac{FP}{FP + TN}$ | the proportion of false positives in the total of negative observations |
| False negative rate (FNR) | $FNR = \dfrac{FN}{TP + FN}$ | the proportion of false negatives in the total of positive observations |

565





**Table 4.** GOF index values, AI, DP2D, SI and TSS, for the calibration run using DTM1 and taking into account the effect of forest.

| calibration run | FPR | TPR | AI | D2PC | SI | TSS |
|---|---|---|---|---|---|---|
| **01** (rg 0.15, 0.40, 0.80) | 0.9932 | 0.0051 | **0.9941** | **0.0085** | 0.9304 | 0.9881 |
| **02** (rg 0.16, 0.41, 0.81) | 0.9834 | 0.0045 | 0.9895 | 0.0172 | 0.9335 | 0.9789 |
| **03** (rg 0.17, 0.42, 0.82) | 0.9702 | 0.0038 | 0.9832 | 0.0301 | 0.9367 | 0.9663 |
| **04** (rg 0.18, 0.43, 0.83) | 0.9574 | 0.0034 | 0.9770 | 0.0427 | **0.9376** | 0.9540 |
| **05** (rg 0.19, 0.44, 0.84) | 0.9411 | 0.0031 | 0.9690 | 0.0590 | 0.9366 | **0.9380** |
| **06** (rg 0.20, 0.45, 0.85) | 0.9186 | 0.0029 | 0.9579 | 0.0814 | 0.9332 | 0.9158 |
| **07** (rg 0.21, 0.46, 0.86) | 0.8927 | 0.0026 | 0.9451 | 0.1073 | 0.9290 | 0.8901 |
| **08** (rg 0.22, 0.47, 0.87) | 0.8739 | 0.0024 | 0.9358 | 0.1261 | 0.9263 | 0.8715 |
| **09** (rg 0.23, 0.48, 0.88) | 0.8551 | 0.0023 | 0.9264 | 0.1450 | 0.9231 | 0.8528 |
| **10** (rg 0.24, 0.49, 0.89) | 0.8347 | 0.0021 | 0.9163 | 0.1653 | 0.9188 | 0.8325 |
| **11** (rg 0.25, 0.50, 0.90) | 0.8193 | 0.0020 | 0.9086 | 0.1807 | 0.9166 | 0.8173 |
| **12** (rg 0.26, 0.51, 0.91) | 0.8039 | 0.0019 | 0.9010 | 0.1962 | 0.9139 | 0.8020 |

570 **Table 5.** Validation results for modelling scenarios taking into account different spatial resolutions and two modelling scenarios: with and without the effect of forest.

| | SCENARIO WITHOUT FOREST | | | | SCENARIO WITH FOREST | | | |
|---|---|---|---|---|---|---|---|---|
| spatial resolution | sensitivity | specificity | FPR | FNR | sensitivity | specificity | FPR | FNR |
| **DTM1** | 0.999 | 0.990 | 0.010 | 0.001 | 0.993 | 0.995 | 0.005 | 0.007 |
| **DTM2** | 0.975 | 0.992 | 0.008 | 0.025 | 0.936 | 0.996 | 0.004 | 0.064 |
| **DTM5** | 0.937 | 0.995 | 0.005 | 0.063 | 0.905 | 0.997 | 0.003 | 0.095 |
| **DTM10** | 0.984 | 0.995 | 0.005 | 0.016 | 0.979 | 0.996 | 0.004 | 0.021 |
| **DTM12.5** | 0.927 | 0.997 | 0.003 | 0.073 | 0.927 | 0.997 | 0.003 | 0.073 |
| **DTM25** | 0.969 | 0.997 | 0.003 | 0.031 | 0.969 | 0.997 | 0.003 | 0.031 |

575





**Table 6.** The maximum values of output rasters of the RockyFor3D model for scenarios with and without forest: the mean of the maximum kinetic energy in kJ (E_mean), the maximum energy value recorded in a given cell/the 95% confidence interval of all maximum kinetic energy values in each cell in kJ (Ph_95CI), the number of blocks stopped in each cell (Nr_d), and the number of block passes through each cell (Nr_p).

| spatial resolution | WITHOUT FOREST | | | | WITH FOREST | | | |
|---|---|---|---|---|---|---|---|---|
| | E_mean | Ph_95CI | Nr_d | Nr_p | E_mean | Ph_95CI | Nr_d | Nr_p |
| DTM1 | 3434.2 | 23322.4 | 16910 | 290739 | 3562.5 | 14206.5 | 19058 | 272416 |
| DTM2 | 3035.1 | 10303.3 | 4433 | 109626 | 2598.0 | 4303.2 | 4307 | 97640 |
| DTM5 | 1527.1 | 2340.05 | 3538 | 21954 | 1537.6 | 2386.6 | 3598 | 21015 |
| DTM10 | 1483.9 | 2297.9 | 2463 | 11168 | 1495.8 | 2299.6 | 2481 | 10904 |
| DTM12_5 | 1388.9 | 2324.4 | 2091 | 6967 | 1534.2 | 2331.0 | 2153 | 6959 |
| DTM25 | 1366.0 | 2209.8 | 892 | 2020 | 1364.0 | 2211.2 | 907 | 2002 |

580

**Table 7.** The number of trees in the propagation area that changes when modelling with the effect of forest (Nr_trees), average DBH values of those trees (DBH average) and the mean of the maximum kinetic energies (E_mean) reached at the locations of those trees.

| spatial resolution | Nr_trees | DBH_average | E_mean |
|---|---|---|---|
| DTM1 | 254 | 39.17 | 337.67 |
| DTM2 | 255 | 39.39 | 407.17 |
| DTM5 | 116 | 35.96 | 354.22 |
| DTM10 | 44 | 29.15 | 212.40 |
| DTM12_5 | 92 | 41.81 | 178.85 |
| DTM25 | 77 | 28.47 | 95.70 |