# Peer review of "Rockfall modelling in forested areas: the role of digital terrain model spatial resolution"

_Natural Hazards and Earth System Sciences, 2019_

## Referee Comment (RC1) · Anonymous Referee #1 · 20 Nov 2019

This contribution is a sensitivity analysis of the code Rockyfor3D for various DEM grid cell sizes, and forest vs no-forest cover. A case study in Slovenia is used to calibrate / compare the simulations. Several indexes of "goodness of fit" are used to compare the actual propagation area to the modeled ones (in binary mode).

In the present form, this paper is more a technical report about some parametric tests made with Rockyfor3D than a research paper. As Rockyfor3D is widely used in the rockfall community, some results may be of interest, however not bringing a significant contribution to the understanding of rockfall processes or even to this specific numerical model.

Some important points are missing: - There is no proper description of the site, no profile of the slope, no indication of the source area, and a map of the soil types. Figure

1 is of very bad quality, with a large map of Eastern Europe and a small "unreadable" picture of the site. - There is no link between Rockyfor3D parameters and rockfall physics. One could expect some in the method or discussion parts. The relationships between soil type, restitution coefficients and rg coefficient are not discussed. A soil type = 3 is selected for the whole area (including the source ?), but the Rg coefficients seem to better correspond to a type 4. However, as the site is not really described, it's not possible for the reader to have a clear view on that. Discussion is also limited to "what if" questions (what if DEM resolution increases...), but no explanation are provided about the reasons / mechanisms. The relation (ratio) between the block size and rg coefficients is a critical point in Rockyfor3D and should have been discussed. - There is a so-called calibration procedure of rg coefficient done on the 1m resolution DEM (even if Rockyfor3D was not made to work with such high resolutions). There is no explanation about the range of values selected. Finally, the best set of parameters corresponds to the smallest values of the range. This is NOT an optimization (line 270) and we don't understand why not trying smaller values.

For all these considerations, I would not recommend this contribution to be published in NHESS. Even if people working specifically with Rockyfor3D may find some hints, the overall scientific content is too poor for a scientific journal.
* * *

---

## Referee Comment (RC2) · Thierry Oppikofer (Referee) · 27 Feb 2020

General comments

The proposed topic is interesting and relevant for rockfall hazard assessment. The spatial resolution of the digital terrain model (DTM) used in all numerical simulations is critical and a good trade-off needs to be found between best possible resolution and computation time, but also possible artefacts of too small DTM cell sizes. The authors calibrate the Rockyfor3D rockfall model parameters with a 1-m DTM using a past rockfall that occurred in 2017 and had a quite large volume (nearly 30'000 m$^3$). They the vary the DTM cell size from 1 to 25 and assess the effect of spatial resolution on the modelled run-out distance and area in comparison with the true extent of the

2017 rockfall. This approach is sound, but the data analysis and interpretation have many major flaws that need to be addressed prior to publication in NHESS.

Specific issues

1. I have doubts that the large volume of the 2017 rockfall is appropriate for a study with Rockyfor3D, as the model is more intended for fragmental rockfalls (single blocks) instead of large volumes that fragment during the event. Using a scree slope formed by multiple rockfall events might provide a more realistic test site. The study would anyway also benefit from several test sites in order to gain more substantial conclusions.

2. The study lacks details on the method used to locate the trees (using the FInT tool provided with Rockyfor3d?) and how the tree locations have been adapted at different spatial resolutions. If the tree location file created by FInT for the 1-m DTM is used for all simulations, I do not understand why there would be so significant differences in number of trees, tree diameter and kinetic energy as shown in Table 7. If FInT is used with different spatial resolutions this should be explained and differences should be shown on maps. In order to provide a complete study of the topic, the authours should also test the alternative approach in Rockyfor3D with the raster files containing tree density and tree diameter etc. How is this approach affected by changes in DTM resolution?

3. Regarding the rockfall model parameters, there are several problems in the calibration:

- The best-fit surface roughness (Rg) parameters are the smallest ones (calibration run 01), while higher values yield poorer results. Please test also with Rg values smaller than those used in calibration run 01, i.e. until the values give worse results. Like that you tend to the real optimum parameter set.

- The initial fall height is set to 50 m which seems excessive considering that you have high-resolution LiDAR data that should correctly depict the location of the rockfall
source area and thus of the height difference between the source area and the toe of the cliff. The additional fall height of 50 m is probably the reason why the run-out area is always overestimated.

- Using a variation of the rockfall dimensions by 50% is appropriate for hazard assessment as it expresses the spread in rockfall volumes observed in the field. For this study, I would however use only a fixed rockfall volume (0% variation) in order to focus the test only on the effect of DTM spatial resolution and on the forest.

4. The goodness-of-fit indices and modelling accuracy statistics need to be used more appropriately and carefully:

- The authors use many different statistics to assess the goodness-of-fit between modelled and observed run-out areas. The authors should select fewer indices, as many of them are related to each other and the reader gets lost. The sensitivity and the false negative rate always sum up to 1 (idem for the specificity and the false positive rate). The whole second paragraph in section 3.2 is therefore redundant with the first paragraph in section 3.2!

- The indices TPR and FPR are used in the text and in Table 4, but they are not defined in the text or tables.

- In Table 4, the headings FPR and TPR cannot be correct. The best TSS value is also obtained for calibration run 01. Based on those results, I suppose that the calculation of the SI cannot be correct and should also be best for run 01.

- Regarding the results of the changing spatial resolution (Table 5), the results should be corrected for the change in cell size and how to attribute cells that are partially in the real run-out area and partially out of it. Attributing the whole cell to the TP or FP might lead to false results; taking instead the exact area located inside or outside will likely be more correct. This effect amplifies with larger DTM cell sizes.

5. The entire section 3.3 on the comparison of model outputs with and without forest needs to be refocused and corrected. Many of the statements in the text are in disagreement with Tables 6 and 7. Furthermore, Table 6 presents several errors:

- It is unclear how the E_mean, Ph_95CI, Nr_d, Nr_p parameters are computed. Is it for the entire modelled run-out area or only for the cells located within the observed run-out area? I would rather use a fictive rockfall fences (or screens) in the central and distal parts of the observed run-out area in order to assess the number of blocs, their energy and passage height at those screens, and use those results in order to assess the effect of forest and spatial resolution.

- The Ph results in Rockyfor3D are usually the passage height and not the maximum kinetic energy, but values provided in Table 6 cannot be the passage height.

- The number of blocks deposited and number of blocs passing through a cell need to be corrected for the total number of simulated rockfalls. With larger DTM cell sizes you have fewer source cells and thus a smaller number of total simulated blocs, which should explain most of the differences observed in the number of passing and deposited blocks.

6. All analyses and interpretations need to be checked again in light of above comments and the entire discussion and conclusion section needs to be reworked. The present conclusions seem not relevant enough for publication in NHESS.

Technical corrections

- The use of the term "DTM spatial resolution" can be somewhat misleading when writing about "better resolution" (=smaller cell size), "increasing resolution" (=smaller cell size) or "decreasing resolution" (= larger cell size). Using "DTM cell size" instead of "DTM spatial resolution" avoids this ambiguity.

- How did you resample the DTM for larger cell sizes? A raster aggregation function with the median elevation value is generally recommended. Using a resolution of 12.5 m might be problematic as it is not an entire multiple of the original resolution, which

likely leads to resampling artefacts.

- Most numbers are given with too high precision (e.g. line 84: area of 19,342 m$^2$, whole section 3.3, Tables 4, 6 and 7), especially when considering the uncertainties in the modelling –> reduce to 3 significant digits.

- Cited references: The references should be more focused on the intended point they refer to. One general reference would suffice for example for the description of the rockfall phenomena (line 32) (Petje et al. 2006 and Lopez-Saez et al. 2016 are well not the first to describe the phenomenon of rockfalls). More pertinent references could also be given for other statements in the introduction (lines 34, 37, 40).

- Lines 64-65: It would be interesting to summarize the findings of other studies focusing on DTM resolution and compare them to your findings.

- Line 98: the definition of the maximum kinetic energy is too vague. Later you use the 95% confidence interval, but also the mean of the maximum kinetic energy. –> specify what is what...

- Line 115: there is a mismatch in the size of the 2017 event (4000 m$^2$ here against 19,342 m$^2$ in line 84)

- Lines 190-192: Explain why the sensitivity is higher in models without forest, while the specificity is higher in models with forest. This seems contradictory and needs thus explanation.

- Line 211: explain where the underestimation occurs (in the SW) and explain why the underestimation occurs there (morphology etc.)

- Table 3: correct the formula of the specificity (FP instead of TP in the denominator), provide also the range of values and optimal value (as in Table 2)

- Figure 1: add a local map of the study area, a field photograph and provide the dimensions of the rockfall in the aerial image.
- Figure 4 cannot be correct. I suspect that the graphs depict the number of cells and not the area (multiply the number of cells by the square of the cell size)

[Figure]

---

## Author Comment (AC1) · 9 Apr 2020

**REFEREE 1**

*This contribution is a sensitivity analysis of the code Rockyfor3D for various DEM grid cell sizes, and forest vs no-forest cover. A case study in Slovenia is used to calibrate /compare the simulations. Several indexes of "goodness of fit" are used to compare the actual propagation area to the modeled ones (in binary mode).*

*In the present form, this paper is more a technical report about some parametric tests made with Rockyfor3D than a research paper. As Rockyfor3D is widely used in the rockfall community, some results may be of interest, however not bringing a significant contribution to the understanding of rockfall processes or even to this specific numerical model.*

The main goal of the submitted paper was certainly not to show how to do parametric tests with the widely used Rockyfor3D model, however some tests were used in order to show the possible calibration of the model in a real case study. Therefore, our purpose was not to just apply some model parameter values "blindly", but to justify the decision on why certain parameter values were used for further simulations. Due to the several (similar) calibration runs and the table summarising results, we understand that it might partially come across as a technical test of the model, however we think that is not the case. To minimise this impression of a technical paper, we will rewrite and rearrange this part in the revised manuscript, by reducing the length of this part in the revised manuscript, improving the graphical support of the results of calibration, and reducing the number of GOF indices to the two (sensitivity and specificity) most appropriate ones. In this way, the flow of the revised manuscript will be redirected to the topic that was really the main subject of study, namely to the influence of grid cell size of DTM and protection effect of forest in rockfall modelling.

*Some important points are missing:*
*There is no proper description of the site, no profile of the slope, no indication of the source area, and a map of the soil types. Figure 1 is of very bad quality, with a large map of Eastern Europe and a small "unreadable" picture of the site.*

The revised manuscript will include a more detailed description of the study site along with the improved map of the study area (a map showing the source area and soil types). The orthophoto image on the map will be increased, we will provide a photo from the field with the metrics of the rockfall (profile of the slope), and we believe that improved map will improve the visualisation of the studied rockfall.

*There is no link between Rockyfor3D parameters and rockfall physics. One could expect some in the method or discussion parts. The relationships between soil type, restitution coefficients and rg coefficient are not discussed. A soil type = 3 is selected for the whole area (including the source ?), but the Rg coefficients seem to better correspond to a type 4. However, as the site is not really described, it's not possible for the reader to have a clear view on that. Discussion is also limited to "what if" questions (what if DEM resolution increases: : :), but no explanation are provided about the reasons/mechanisms. The relation (ratio) between the block size and rg coefficients is a critical point in Rockyfor3D and should have been discussed.*

As argued by the Referee, we agree that there is a lack of presentation/discussion about the relationship between soil type, restitution coefficient and rg coefficients. Therefore, in the revised manuscript we will focus on this relationship, both in improving the methodological part as well as the discussion. In order to justify the chosen soil types, we will make an

additional mapping of soil types in the field and we will divide the potential runout area in several soil types based on these additional field observations. Thorough explanation of the chosen soil types in the propagation and runout area will be given along with the photograph evidence for better visualisation and description of the study area. In the current manuscript version we used soil type 4 - talus slope (Ø > ~ 10 cm or compacted soil with large rock fragments) for the whole potential propagation and runout area, excluding the source area which is in the model defined separately. We agree that the discussion needs to be rewritten into more detail, especially considering the relation between soil type and restitution coefficient - rg coefficients, and their effect on modelling results using DTMs with different grid cell sizes. In addition, as our main focus was on the protective role of forest, more conclusions on this side will be provided in order to give readers some outcomes that would be useful when modelling rockfalls in forest, and on different spatial scales.

*There is a so-called calibration procedure of rg coefficient done on the 1m resolution DEM (even if Rockyfor3D was not made to work with such high resolutions). There is no explanation about the range of values selected. Finally, the best set of parameters corresponds to the smallest values of the range. This is NOT an optimization (line 270) and we don't understand why not trying smaller values.*

Since the authors of the model have expressed that the optimal DTM grid cell size lies between 2 and 10m, we agree that 1m resolution shouldn't have been used in the calibration stage. Regarding the calibration on 1m, we have decided to test it on the best possible resolution (i.e. 1m), however, in the revised manuscript we will use grid cell size of 2m for the calibration procedure. The justification of why not more rg values were not used: we have used an arbitrary interval of rg values that were used which lead to the state where the smallest values correspond to the most successful ones, and yes, we should have continued with the calibration procedure with even lower rg values. In the revised manuscript we will not use the same increment of 0.01 in changing rg values, but we will use randomly selected values from intervals (e.g. 30 combinations), i.e. rg70 between 0.05 and 0.30, rg20 between 0.30 and 0.7, and rg10 between 0.7 and 0.9 – without overlapping of intervals. By such computations we will get an optimum combination of rg values. In such way we will be able to observe how the length and area of the runout area will change, and where there is a limit between the more or less successful modelling of rockfall propagation and runout area.

*For all these considerations, I would not recommend this contribution to be published in NHESS. Even if people working specifically with Rockyfor3D may find some hints, the overall scientific content is too poor for a scientific journal.*

We will consider above written comments provided by the Referee, and with that we believe that we will improve its scientific value as a research paper. The major changes to the manuscript will be: i) decreasing the content (e.g. tables, GOF indices) on the calibration so that the paper will not be read as a technical report, ii) improving the presentation map of the study site, including additional photographs of the area and metrics of the rockfall, iv) improving the categorization of the study site in different, more precise soil types based on the additional field collection of data, also leading to improved discussion in that relation accordingly to the DTMs grid cell sizes, v) using DTM grid cell of 2m in calibration stage instead of 1m, and vi) using wider set of values of rg coefficients in the calibration stage, not only changing them in the linear way but also randomly with all three rg coefficients (rg70, rg20, rg10). Furthermore, considering also comments of the Referee #2, we will add to the

revised manuscript another case study of a nearby rockfall triggered recently that will add further discussion on the application possibilities of the used rockfall model.

---

## Author Comment (AC2) · 9 Apr 2020

**REFEREE 2**

*GENERAL COMMENTS*

*The proposed topic is interesting and relevant for rockfall hazard assessment. The spatial resolution of the digital terrain model (DTM) used in all numerical simulations is critical and a good trade-off needs to be found between best possible resolution and computation time, but also possible artefacts of too small DTM cell sizes. The authors calibrate the Rockyfor3D rockfall model parameters with a 1-m DTM using a past rockfall that occurred in 2017 and had a quite large volume (nearly 30'000 m³). They the vary the DTM cell size from 1 to 25 and assess the effect of spatial resolution on the modelled run-out distance and area in comparison with the true extent of the 2017 rockfall. This approach is sound, but the data analysis and interpretation have many major flaws that need to be addressed prior to publication in NHESS.*

*SPECIFIC ISSUES*

*1. I have doubts that the large volume of the 2017 rockfall is appropriate for a study with Rockyfor3D, as the model is more intended for fragmental rockfalls (single blocks) instead of large volumes that fragment during the event. Using a scree slope formed by multiple rockfall events might provide a more realistic test site. The study would anyway also benefit from several test sites in order to gain more substantial conclusions.*

The study site that was studied in the manuscript was chosen since: i) it is located within/above forest and hence we can study the impact of forest on rockfall dynamics, and ii) rockfall dynamics did correspond to a situation of fragmental blocks that are travelling individually down the slope. Those rocks actually travelled the longest distances and also in these situations the forest had an impact on reducing their trajectories. Therefore, we consider that Rockyfor3D is suitable for simulation of the studied rockfall. In order to provide more substantial conclusions, we will include additional test site in the revised manuscript. The additional rockfall has just occurred recently (in March 2020), it is located on the opposite slope in the same valley, it's of lower volume (approximately 11,000 m³), and the forest had an important impact on stopping individual rock blocks.

*2. The study lacks details on the method used to locate the trees (using the FInT tool provided with Rockyfor3d?) and how the tree locations have been adapted at different spatial resolutions. If the tree location file created by FInT for the 1-m DTM is used for all simulations, I do not understand why there would be so significant differences in number of trees, tree diameter and kinetic energy as shown in Table 7. If FInT is used with different spatial resolutions this should be explained and differences should be shown on maps. In order to provide a complete study of the topic, the authors should also test the alternative approach in Rockyfor3D with the raster files containing tree density and tree diameter etc. How is this approach affected by changes in DTM resolution?*

The effect of forest was calculated based on a tree file containing the locations of the trees and their DBH, which were extracted based on the 1m lidar data. The same tree file was used for all simulations using different DTMs. The authors of the model provide an additional tool (FINT) that enables one to extract the locations of the trees. The calculation is based on the digital surface model and digital terrain model. The tool provides the locations of individual trees, and their diameter at breast height (DBH), using a standard or customize function based on the tree height curve. Since its functionality was not presented in the current version of the

manuscript, we will include it to the revised manuscript. Delamination of the trees with this tool can only be done with data with DTM grid cell size up to 5m since resolutions with larger grid cell sizes do not allow a correct extraction of individual trees. Also, as we want to have the most realistic locations of the trees and their spatial distributions, we wouldn't be able to extract the locations of all trees with other grid cell sizes. Even more, if we did a field survey of the trees, the final input into the model would be the same with all grid cell sizes. The main difference will then be later in the calculations where more trees are located within one raster cell. This is the reason why we used 1m resolution to extract the trees and also why we used it in calculations with all DTM grid cell sizes. The reason why we didn't use the forest raster maps is that they are less accurate as they only provide average value of number of trees and their DBH, and would in our case be extracted from the forest maps that are of lower quality compared to lidar data. The estimated density of trees based on these maps is a rough estimate, and we would in any case be using the data on number of trees, extracted from the lidar data. Due to this fact the modelling outcome would not differ between the two forest options in the model.

The actual number of trees that was input into the model was the same with all cases. With the number of trees shown in Table 7 we wanted to indicate how many trees are actually located within the modelled propagation area, and with that how did forest actually contribute in reducing the runout area – less trees means larger runout area. In Table 7, it is possible to observe that with larger cell size the propagation area is reducing (due to larger grid cell sizes), not reaching as many trees in the runout area that would be able to reduce the kinetic energy of the rocks in the simulation. Due to that fact the propagation area of DTM10, DTM12.5 and DTM25 doesn't differ significantly when forest is or isn't included into the model.

To improve the calculation of tree DBH's which is in the model relevant in stopping rocks, we will additionally measure DBH and height of representative sample of trees at both locations to be included in the revised manuscript. Based on the measurements we will be able to provide a customize function to FINT tool for calculation of tree DBH that will be specific for each rockfall location, leading to more realistic results that will not strictly rely on already defined calculations in the FINT model.

*3. Regarding the rockfall model parameters, there are several problems in the calibration:*

*- The best-fit surface roughness (Rg) parameters are the smallest ones (calibration run 01), while higher values yield poorer results. Please test also with Rg values smaller than those used in calibration run 01, i.e. until the values give worse results. Like that you tend to the real optimum parameter set.*

We have discussed this issue already when answering to the Referee #1. We have used an arbitrary interval of rg values that lead to the state where the smallest values correspond to the most successful one, and we should have continued with the calibration with even lower rg values. In the revised manuscript we will not use the same increment of 0.01 in changing rg values, but we will use randomly selected values from intervals (e.g. 30 combinations), i.e. rg70 between 0.05 and 0.30, rg20 between 0.30 and 0.7, and rg10 between 0.7 and 0.9 – without overlapping of intervals. By such computations we will get an optimum combination of rg values. In such way we will be able to observe how the length and area of the runout area will change, and where there is a limit between the more or less successful modelling of rockfall propagation and runout area.

*- The initial fall height is set to 50 m which seems excessive considering that you have high-resolution LiDAR data that should correctly depict the location of the rockfall source area and thus of the height difference between the source area and the toe of the cliff. The additional fall height of 50 m is probably the reason why the run-out area is always overestimated.*

The initial fall height was set at 50 meters, since it was measured based on the change in elevation between the source area and the toe of the cliff, using lidar data. Some parts of the cliff are even higher; therefore we believe that the initial fall height should not be changed. However, we will for the model sensitivity purposes include also a few different fall heights in order to observe how the fall height impact the rockfall runout area and the rock rebound heights. With that we will be able to see the model sensitivity considering different fall heights and rg values, especially to see if the initial fall height can in some cases be reduced as suggested by the Referee.

*- Using a variation of the rockfall dimensions by 50% is appropriate for hazard assessment as it expresses the spread in rockfall volumes observed in the field. For this study, I would however use only a fixed rockfall volume (0% variation) in order to focus the test only on the effect of DTM spatial resolution and on the forest.*

In the model we have used the variation of the volume by 50 % as we observed it in the field. Since it is our main purpose to study the effect of DTM grid cell size and the effect of forest on modelling rockfall propagation and runout areas, we agree with the Referee that the variation of volume should be set to 0 %, and we will take this into account in the revised manuscript. For the sake of discussion we will take a few examples at the maximum extent of the rockfall and compare the differences when simulating different rockfall volumes to observe its impact on the rockfall runout area.

*4. The goodness-of-fit indices and modelling accuracy statistics need to be used more appropriately and carefully:*

*- The authors use many different statistics to assess the goodness-of-fit between modelled and observed run-out areas. The authors should select fewer indices, as many of them are related to each other and the reader gets lost. The sensitivity and the false negative rate always sum up to 1 (idem for the specificity and the false positive rate). The whole second paragraph in section 3.2 is therefore redundant with the first paragraph in section 3.2!*

We agree with the suggestion of the Referee to reduce the number of goodness-of-fit indices. In the revised version we will only use two - sensitivity and specificity. Consequently, we will reduce the amount of text regarding this topic and delete the content that is being repeated.

*- The indices TPR and FPR are used in the text and in Table 4, but they are not defined in the text or tables.*

The indices TPR and FPR are defined in Table 3.

*- In Table 4, the headings FPR and TPR cannot be correct. The best TSS value is also obtained for calibration run 01. Based on those results, I suppose that the calculation of the SI cannot be correct and should also be best for run 01.*

In the Table 4 the first column should have been TPR and the second FPR, for what we apologize. As observed, the best values should have been in the case of TSS and SI with calibration run 01. This will be corrected in the revised manuscript, where also the number of GOF indices will be reduced to two (sensitivity and specificity).

*- Regarding the results of the changing spatial resolution (Table 5), the results should be corrected for the change in cell size and how to attribute cells that are partially in the real run-out area and partially out of it. Attributing the whole cell to the TP or FP might lead to false results; taking instead the exact area located inside or outside will likely be more correct. This effect amplifies with larger DTM cell sizes.*

The effect of changing grid cell size and how to attribute cells could be partially in the real runout area and partially out of it occurs with rasterization of vector file. However, all goodness-of-fit indices used in this manuscript use the raster analysis, where raster cells are either categorized as TP, TN, FP or FN. Each cell can only have one value. In this case we had to rasterize the real (vectorised) rockfall extent, but we did it for each grid cell size separately, and when rasterizing this issue cannot occur that one cell would be partially in or out. Clearly with larger raster cell the generalization will be larger thus leading to a higher error which is also the main topic that we wanted to show in this manuscript.

*5. The entire section 3.3 on the comparison of model outputs with and without forest needs to be refocused and corrected. Many of the statements in the text are in disagreement with Tables 6 and 7. Furthermore, Table 6 presents several errors:*

*- It is unclear how the E_mean, Ph_95CI, Nr_d, Nr_p parameters are computed. Is it for the entire modelled run-out area or only for the cells located within the observed run-out area? I would rather use a fictive rockfall fences (or screens) in the central and distal parts of the observed run-out area in order to assess the number of blocs, their energy and passage height at those screens, and use those results in order to assess the effect of forest and spatial resolution.*

The E_mean, Ph_95CI, Nr_d, and Nr_p are calculated for the whole modelled runout area. The proposed solution for better interpretation of the results is adequate and we will change the manuscript as proposed by the Referee. We will use fictive fences in both central and distal parts of the observed runout area so that we will be able to observe the number of blocks, their energy and passage height, and evaluate it between DTMs with different grid cell size. The results will be shown in two separate but parallel longitudinal profiles along the studied area.

*- The Ph results in Rockyfor3D are usually the passage height and not the maximum kinetic energy, but values provided in Table 6 cannot be the passage height.*

In the Table 6 there was a wrong naming – instead of Ph it should have been E_95CI, and it represents the 95 % confidence interval (CI) of all maximum kinetic energy values (in kJ) in a cell. The mistake will be omitted.

*- The number of blocks deposited and number of blocs passing through a cell need to be corrected for the total number of simulated rockfalls. With larger DTM cell sizes you have fewer source cells and thus a smaller number of total simulated blocs, which should explain most of the differences observed in the number of passing and deposited blocks.*

The number of blocks deposited and passed through each cell will be recalculated for the total number of simulated rockfalls (given in % of the released number of blocks), and the differences between DTMs are due the different cell sizes in the source area. This will give a better comparison between different DTMs used for simulation.

*6. All analyses and interpretations need to be checked again in light of above comments and the entire discussion and conclusion section needs to be reworked. The present conclusions seem not relevant enough for publication in NHESS.*

The analyses and interpretation will be checked again following the comments provided by the Referee. Manuscript will be reworked with major changes which will be: i) adding additional test site, ii) description of the tool used for extraction of tree locations and field collection of data for providing site-specific DBH function, iii) improving calibration of the rg coefficients – using the wider extent of values and also changing their values in a non-linear way and randomly with all three rg coefficients (rg70, rg20, rg10), iv) testing the model with lower fall heights, v) reducing the number of goodness-to-fit indices to only two, vi) exclusion of DTM 12.5m and 25m from the analyses, and vii) using cell size 2m for calibration of the model. Changes in the initial manuscript will be relevant and will bring additional data that will be able to enrich both the result and discussion part of the manuscript. We will devote more content to explaining why there are differences between grid cell sizes, and accordingly also how the protection effect of forest against rockfall is taken into account. Furthermore, we will add analysis of a second rockfall triggered recently nearby in the same valley, adding more content for the discussion on the effect of forest regarding the extents of rockfall events, and possibilities of application of the rockfall model.

**TECHNICAL CORRECTIONS**

*- The use of the term "DTM spatial resolution" can be somewhat misleading when writing about "better resolution" (=smaller cell size), "increasing resolution" (=smaller cell size) or "decreasing resolution" (= larger cell size). Using "DTM cell size" instead of "DTM spatial resolution" avoids this ambiguity.*

The use of the term "increasing/decreasing spatial resolution" will be replaced with the term "DTM grid cell size" as proposed. Accordingly, the title of the manuscript will be changed by replacing spatial resolution with the grid cell size.

*- How did you resample the DTM for larger cell sizes? A raster aggregation function with the median elevation value is generally recommended. Using a resolution of 12.5 m might be problematic as it is not an entire multiple of the original resolution, which likely leads to resampling artefacts.*

DTM's for different grid cell sizes were created based on the lidar point cloud, using the binning interpolation type with average elevation values. Since the use of DTMs with grid cell size larger than 10m is not reasonable on local scale and because model is intended for the use for resolutions between 2 and 10m, and in order to avoid the possible artefacts in DTMs, we will exclude 12.5m and 25m in revised version of the manuscript. Since optimal grid cell size for the model lies between 2 and 10m we will also use grid cell size of 2m in the calibration process.

*- Most numbers are given with too high precision (e.g. line 84: area of 19,342 m², whole section 3.3, Tables 4, 6 and 7), especially when considering the uncertainties in the modelling –> reduce to 3 significant digits.*

The number of significant digits will be reduced.

*- Cited references: The references should be more focused on the intended point they refer to. One general reference would suffice for example for the description of the rockfall phenomena (line 32) (Petje et al. 2006 and Lopez-Saez et al. 2016 are well not the first to describe the phenomenon of rockfalls). More pertinent references could also be given for other statements in the introduction (lines 34, 37, 40).*

The references in the Introduction part of the manuscript will be improved, including more relevant (often cited) and suitable references.

*- Lines 64-65: It would be interesting to summarize the findings of other studies focusing on DTM resolution and compare them to your findings.*

Short summary of other studies focusing on the DTM resolution will be added to the Introduction part of the manuscript, and comparison to our study will be included in the Discussion part.

*- Line 98: the definition of the maximum kinetic energy is too vague. Later you use the 95% confidence interval, but also the mean of the maximum kinetic energy. –> specify what is what…*

A more detailed definition of maximum kinetic energy will be added.

*- Line 115: there is a mismatch in the size of the 2017 event (4000 m² here against 19,342 m² in line 84)*

In the line 84 it should have been different units (m³) – we will correct it.

*- Lines 190-192: Explain why the sensitivity is higher in models without forest, while the specificity is higher in models with forest. This seems contradictory and needs thus explanation.*

The sensitivity is higher in models without forest since this model overestimates the actual extent of rockfall meaning that more raster cells will be classified as TP. However, the specificity will be lower due to this fact with this model and therefore the specificity will be higher with the model that considers forest.

*- Line 211: explain where the underestimation occurs (in the SW) and explain why the underestimation occurs there (morphology etc.)*

The underestimation of the propagation area is the lowest in the south-western part – it is the lowest with DTM1 where with forest scenario it almost achieves a perfect fit. Perhaps the reason why is that the terrain in this direction is flatter in longer distance than in the north and north western direction where the runout length in the less rugged surface in shorter. In that part the terrain starts to decent quicker, providing more energy so that the rocks can be moved further.

*- Table 3: correct the formula of the specificity (FP instead of TP in the denominator), provide also the range of values and optimal value (as in Table 2)*

The formula of specificity will be corrected, and the range of optimal values will be provided in the revised manuscript.

*- Figure 1: add a local map of the study area, a field photograph and provide the dimensions of the rockfall in the aerial image.*

A local map of the study area, a field photograph and metrics of the rockfall in the aerial image will be added to the revised manuscript.

*- Figure 4 cannot be correct. I suspect that the graphs depict the number of cells and not the area (multiply the number of cells by the square of the cell size)*

The numbers in the Figure 4 were wrong – the graph will be changed accordingly.